# In Vitro Hemocompatibility and Genotoxicity Evaluation of Dual-Labeled [^99m^Tc]Tc-FITC-Silk Fibroin Nanoparticles for Biomedical Applications

**DOI:** 10.3390/ph16020248

**Published:** 2023-02-07

**Authors:** María Alejandra Asensio Ruiz, Ángela Alonso García, María de la Luz Bravo-Ferrer Moreno, Iria Cebreiros-López, José Antonio Noguera-Velasco, Antonio Abel Lozano-Pérez, Teresa Martínez Martínez

**Affiliations:** 1Servicio de Radiofarmacia, Hospital Clínico Universitario Virgen de la Arrixaca, 30120 Murcia, Spain; 2Instituto Murciano de Investigación Biosanitaria (IMIB)-Arrixaca, 30120 Murcia, Spain; 3Servicio de Análisis Clínicos, Hospital Clínico Universitario Virgen de la Arrixaca, 30120 Murcia, Spain; 4Departamento de Biotecnología Genómica y Mejora Vegetal, Instituto Murciano de Investigación y Desarrollo Agrario y Medioambiental, 30150 Murcia, Spain

**Keywords:** radiolabeling, silk fibroin NPs, ^99m^Tc, FITC, SPECT, nuclear imaging, multimodal imaging, hemocompatibility, genotoxicity, CBMN assay

## Abstract

Nuclear imaging is a highly sensitive and noninvasive imaging technique that has become essential for medical diagnosis. The use of radiolabeled nanomaterials capable of acting as imaging probes has shown rapid development in recent years as a powerful, highly sensitive, and noninvasive tool. In addition, quantitative single-photon emission computed tomography (SPECT) images performed by incorporating radioisotopes into nanoparticles (NPs) might improve the evaluation and the validation of potential clinical treatments. In this work, we present a direct method for [^99m^Tc]Tc-radiolabeling of FITC-tagged silk fibroin nanoparticles (SFN). NPs were characterized by means of dynamic light scattering and scanning electron microscopy. In vitro studies were carried out, including the evaluation of stability in biological media and the evaluation of hemocompatibility and genotoxicity using the cytokinesis block micronucleus (CBMN) assay. The radiolabeling method was reproducible and robust with high radiolabeling efficiency (∼95%) and high stability in biological media. Hydrodynamic properties of the radiolabeled NPs remain stable after dual labeling. The interaction of SFN with blood elicits a mild host response, as expected. Furthermore, CBMN assay did not show genotoxicity induced by [^99m^Tc]Tc-FITC-SFN under the described conditions. In conclusion, a feasible and robust dual-labeling method has been developed whose applicability has been demonstrated in vitro, showing its value for further investigations of silk fibroin NPs biodistribution in vivo.

## 1. Introduction

In the last few decades, the application of nanotechnology in the field of medicine and health has increased exponentially [1]. In this area, new and innovative formulations containing radioisotopes with several unique applications in imaging, diagnostics, targeted cancer therapy, drug delivery, and theragnostic purposes have emerged, transforming current personalized nanomedicine approaches [1,2].

Nanoparticles (NPs) are versatile candidates as imaging probes in multimodal imaging and theragnosis. Nanomaterials can be applied as probes for various noninvasive imaging modalities, including optical imaging (OI) [3], magnetic resonance imaging (MRI) [4], computed tomography (CT) [5], and nuclear imaging (single-photon emission computed tomography (SPECT) and positron emission tomography (PET)) [6]. Nuclear imaging provides functional and metabolic information at the molecular level through the administration of a radiolabeled probe with a suitable radionuclide. While isotopes such as fluorine-18 (^18^F), gallium-68 (^68^Ga), and copper-64 (^64^Cu) are indicated for PET, technetium-99m (^99m^Tc), indium-111 (^111^In), or gallium-67 (^67^Ga) are used for SPECT. These techniques are well suited for imaging biodistribution in vivo compared to other modalities in terms of high sensitivity (10^−10^–10^−12^ M) [7], deep tissue penetration [8] low toxicity and biocompatibility [9], possibility of quantification in vivo and ex vivo, and high spatial resolution when combined with CT or MRI. Furthermore, the integration of multiple radioisotopes and signal reporters on a single nanoparticle allows for multimodal molecular imaging [10].

NPs are being widely explored in preclinical research; the increasing number of publications of radiolabeled nanomaterials for biomedical applications corroborate the growing interest in the field [2,11]. Nuclear imaging probes, including carbon-based NPs, metal NPs, ceramic NPs, semiconductors (quantum dots), polymeric NPs, and protein-based and lipid-based NPs [7,9,10,12], offer a wide range of physiochemical properties for overcoming handicaps such as fast metabolism and nonspecific biodistribution of small molecules for clinical imaging probes (i.e., ^18^F-fludesoxiglucose). Moreover, the strategies of pharmacokinetic modulations through surface modification for cell internalization, coating for biocompatibility and favorable blood circulation, functional group attachment for enhanced targeting specificity, and the wide loading ability of diagnostic and therapeutic moieties make them a very attractive tool for diagnosis, therapy, and theragnosis [1,9].

Bionanomaterials, including organic and inorganic materials, can be easily assimilated in living systems. These small-sized NPs can penetrate into capillaries and propagate across biological barriers, enabling the detection of changes occurring at molecular levels [9]. Combining the unique properties with being biocompatible would accelerate the application for molecular imaging. Furthermore, understanding cell–nanoparticle interactions is critical not only to develop effective nanosized drug delivery systems [12], but also to predict their biocompatibility, especially in those cases where systemic administration is pursued [13]. In vivo and in vitro studies described cell toxicity after NP exposure depending on type of the material, size, shape, charge, or solubility. Cell toxicity induced by several metallic NPs (Cu, Ag, TiO_2_, ZnO, Au) seems to be mediated by the production of reactive oxygen species (ROS), mitochondrial and lysosomal damage, and mutagenesis [14,15,16], while genotoxicity studies on some polymer-based NPs suggest limited DNA damage [17]. Moreover, biodegradable polymers have properties such as noninflammatory and no-immunogenic nature and in vivo degradation into nontoxic by-products eliminated by normal metabolic pathways [2].

Therefore, natural polymer carriers such as silk fibroin NPs seem to be good candidates to achieve suitable biocompatibility and biodegradation rate, as well as drug loading and release. Silk fibroin has been shown to be an excellent matrix for controlled drug delivery [18], with good biocompatibility and the ability to degrade into harmless products in biological media [10]. The amphipathic nature of the SFN and the presence of reactive amino acid residues (i.e., Asp, Glu, Lys) on the surface enable the labeling of NPs with fluorophores (i.e., FITC) [10,19] and radionuclides such as ^99m^Tc, ^111^In, ^90^Y, ^68^Ga, or ^177^Lu by mechanisms already described [7,20].

Technetium-99m continues being the most widely used radionuclide for SPECT imaging. This radionuclide has excellent physical properties (half-life of 6 h and gamma emission of 140 KeV (89%)) combined with high accessibility based on onsite generator elution, low cost, and the versatility of radiolabeling approaches due to its variety of oxidation states and its metallic behavior. Moreover, in the PET era, ^99m^Tc has continued to evolve, as pointed out by new radiopharmaceuticals development, i.e., [^99m^Tc]Tc-HYNIC-TOC or [^99m^Tc]Tc-PSMA I&S [21,22].

To date, a few ^99m^Tc-labeled NPs, such as ^99m^Tc-labeled colloidal NPs, are clinically approved, and several type of NPs (i.e., iron oxide, gold NPs, micelles, or liposomes) have been under evaluation in preclinical cancer models after ^99m^Tc labeling [2,7].

The aim of this work is to describe the preparation of dual-labeled [^99m^Tc]Tc-FITC-silk fibroin NPs, suitable for in vivo tracking of the SFN by nuclear imaging and complemented with ex vivo fluorescence microscopy. A two-step labeling approach is proposed, wherein NPs are first labeled with FITC, and then with ^99m^Tc by a direct labeling with SnCl_2_ reduction. To complete the in vitro toxicity profile, hemocompatibility and genotoxicity of [^99m^Tc]Tc-FITC-SFN were assessed by means of in vitro blood incubation and cytokinesis block micronucleus assay (CBMN), respectively.

## 2. Results

### 2.1. Radiolabeling Optimization

The radiolabeling procedure was optimized by varying the quantities of SnCl_2_ in HCl 0.33 M (Table 1) added to 1 mg of SFN and ~37 MBq of sodium pertechnetate (Na[^99m^Tc]TcO_4_) in 1 mL of NaCl 0.9% and incubated at room temperature for 10 min.

The quantity of SnCl_2_ added had significant effects on the radiolabeling efficiency (RLE) and the hydrodynamic characteristics of the NPs (diameter (Z_ave_), polydispersity (PDI), and Z potential (ζ)), as can be seen in Table 1. Thus, RLE increased from 88.34 ± 1.11 to 92.13 ± 0.96 when the SnCl_2_ added rose from 7 μg/mL to 20 μg/mL. Moreover, it reached 96.38 ± 0.29 % when 250 μg/mL were added. However, at this concentration, a significant increase in size and a reduction in the absolute value of Z potential (ζ) was observed compared to the control group (*p* < 0.05), while the polydispersity increased gradually with the increment in the concentration.

Additionally, when higher quantities of SnCl_2_ were added (i.e., 500 μg and 1 mg), a considerable increase in the size of the particles took place, appreciating the formation of aggregates. When the concentration of SnCl_2_ reached 1000 μg/mL, the ζ value shifted to ~0 mV and the size of the particles increased to 7350 ± 1143 nm. The zeta potential, which depends on the surface charge, is important for the stability of NPs in suspension [23]. Thus, aggregation of a suspension of NPs is favored when ζ is close to 0 mV [24], which explains the formation of micrometric aggregates.

Once the experimental conditions were optimized, aqueous suspensions of SFN and FITC-SFN were radiolabeled as previously described, achieving statistically different (*p* < 0.0001, Unpaired *t*-test) RLEs of 91.42 ± 0.90% (n = 20) and 98.47 ± 0.66% (n = 20), respectively, as can be seen in Figure 1. The observed differences in the RLE of [^99m^Tc]Tc-SFN and [^99m^Tc]Tc-FITC-SFN indicate that FITC moieties have a positive effect on the radiolabeling of the NPs.

The measured specific activity (SA) was found in the range of 3.38–3.63 MBq/mg (n = 20) and the radiochemical purity (RCP) was found to be 99.04 ± 0.51% for [^99m^Tc]Tc-SFN (n =4) and 98.41 ± 0.85% (n = 4) for [^99m^Tc]Tc-FITC-SFN.

### 2.2. Stability Assays

Both [^99m^Tc]Tc-SFN and [^99m^Tc]Tc-FITC-SFN were stable in saline and in human blood serum, along the time of incubation (4h) according to the DLS analysis, the FESEM images, and the measured percentage of ^99m^Tc eluted from the NPs. Thus, the dual-labeled NPs appear to have good stability either in saline or human blood serum, having only small variations in the hydrodynamic diameter within the measurement uncertainty, as can be seen in Figure 2.

FESEM images of the NPs showed small differences in the morphology of the NPs in solid state after incubation in serum, as can be seen in Figure 3. Images of SFN and FITC-SFN were taken in the same conditions as controls. The NPs were spherical in shape and were well dispersed, showing homogenous size distributions, similar to those previously described in the literature for silk fibroin NPs [10,19]. Although aggregation in solid state of the labeled NPs was clearly visible, silk particles kept their globular morphology after labeling and incubation in serum. It is noteworthy that [^99m^Tc]Tc-FITC-SFN presented a higher aggregation in solid state than [^99m^Tc]Tc-SFN.

In the image correspondent to the [^99m^Tc]Tc-FITC-SFN incubated in serum (Figure 3D), several bright aggregates stand out from the background due their high electrodensity, similar to other [^99m^Tc]Tc-loaded mesoporous silica NPs described previously [25]. This phenomenon can be produced during the dryness of the nanoparticle suspension from the deposition of the “free” ^99m^Tc released from the NPs.

The quantification of the eluted ^99m^Tc from nanoparticles showed a percentage of release in the range of 0.84–2.58% after 4 h of incubation in saline for both types of nanoparticles, confirming that the radionuclide remains attached to the nanoparticles, and as expected, FITC does not affect the stability of the radiolabeling. The incubation with blood serum showed higher percentage of eluted ^99m^Tc (2.75–4.45%), probably related to the processing of the samples. However, no increase in eluted ^99m^Tc was observed along the time of study for [^99m^Tc]Tc-FITC-SFN and [^99m^Tc]Tc-SFN, concluding that radiolabeling is stable in blood serum (See Figure 4).

### 2.3. Cytokinesis Block Micronucleus (CBMN) Assay

The exposition of cells to radiolabeled NPs may produce an increase in the frequency of the nuclear anomalies such as binucleated cells (BNs) with micronuclei (MNi), nuclear buds (NBUDs), and nucleoplasmic bridges (NPBs), according the CBMN assay protocol [26].

DNA damage events were scored specifically as Mni, NBUDs, and NPBs were quantified and expressed as events per 1000 BN (**‰**), and results are shown in Table 2.

Some examples of those BNs observed in cultured in vitro human lymphocytes exposed to the radiolabeled silk NPs are presented in Figure 5.

As can be seen in Table 2, [^99m^Tc]Tc-FITC-SFN did not increase the MNi, NBUDs, or NPBs frequencies compared to FITC-SFN and the negative control (WFI). As expected, FITC-SFNs have no genotoxic effects in conditions described; furthermore, radiolabeling FITC-SFNs with ^99m^Tc does not increase the frequencies of cytotoxicity biomarkers, confirming that the incorporation of the radionuclide in ~37 MBq/mg/mL does not induce DNA damage. The positive control shows an increase in the frequencies as expected, according to the dose of ^99m^Tc added (37 MBq).

### 2.4. Hemocompatibility Assay

Flow cytometry analysis showed a slight reduction in the number of platelets (Figure 6A), red blood cells (Figure 6B), and neutrophils (Figure 6C) after 4 h of incubation in the samples incubated with SFN compared to the controls without NPs (*p* < 0.05, paired *t* test). On the contrary, monocytes, lymphocytes, eosinophils, and basophils did not show changes in their counting after incubation with the NPs.

However, LDH levels were increased in samples incubated with radiolabeled NPs compared to control (*p* < 0.05, paired *t* test), confirming partial cell lysis after 4 h of incubation with SFN (Figure 6D). Stronger response was observed in four of the six donors, where the enzyme values were out of the clinically considered normal range (135–210 U/L). The highest value exceeded 50% the top limit.

Measured values of C-reactive protein (CRP) were in the range considered clinically normal (0.00–0.50 mg/dL) and did not increase after 4 h of incubation with radiolabeled NPs compared to controls (*p* > 0.05, paired *t* test), confirming the absence of inflammatory response to SFN (Figure 6D).

### 2.5. Platelet Aggregation Assay

Platelet aggregation assessed by incubation of the NPs with platelet-rich plasm (PRP) after 30 min of incubation showed percentages of aggregation in the range of 3.0–6.2%, and after 4h slightly increased to values in the range 4.6–12.6%, although without statistical differences (*p* > 0.05, ANOVA). However, positive controls induced aggregation higher than 40%, with values statistically different to the rest of the groups (*p* < 0.05, ANOVA) (see Table 3 for detailed values).

In fact, when PRP was incubated with 0.25 M CaCl_2_ (positive control) the formation of fibrils was observed. On the contrary, when PRP was incubated with SFN, no sign of platelet aggregation was observed (Figure 7).

Activated partial thromboplastin time (APTT) values were in the range considered clinically normal (20–40 s), showing a slight decrease in the values at 30 min (*p* > 0.05, ANOVA) and 4 h (*p* > 0.05, ANOVA) in comparison with the control group, as can be seen in Figure 8A). Prothrombin time (PT) showed values in the reference range (12–15 s), with a slight increase after 30 min of incubation (*p* > 0.05) and 4 h (*p* > 0.05, ANOVA) compared to controls (Figure 8B).

No fibrinolysis was detected along the incubation time, since no differences in D-dimer levels were seen after 30 min (*p* > 0.05, ANOVA) and 4 h (*p* > 0.05, ANOVA) compared to controls, as can be seen in Figure 8C. The same behavior was found in the levels of fibrinogen and albumin, without statistical differences after 4 h of incubation compared to control (*p* > 0.05, paired *t* test), as can be seen in Figure 8D and Figure 8E, respectively.

## 3. Discussion

The development of nanomaterials as nuclear imaging probes has grown rapidly in recent years. Nanoprobes are powerful and highly sensitive tools for in vivo biodistribution studies. Therefore, quantitative SPECT imaging performed by incorporating radioisotopes into NPs can help in the evaluation and validation of potential clinical treatments. Non-chelation-based strategies involve the direct incorporation of radionuclides into the core and/or surface of nanomaterials, avoiding the need for external chelating agents [7]. In our work, ^99m^Tc was incorporated into NPs by chemical adsorption through lowering their oxidation state with stannous chloride; thus, the radionuclide was able to establish coordination bonds with chemical groups on the nanomaterial surface, such as–-SH or –OH in mild conditions. The main drawback of this method is the relatively low strength of the chemical interaction, which results in poor stability of the radiolabel [7]. In this paper, we present a direct approach for [^99m^Tc]Tc-radiolabeling of silk NPs using stannous chloride as a reducing agent in mild conditions, achieving a radiolabeling efficiency of more than 90%, with a specific activity in the range of 3.38-3.63 MBq/mg and radiochemical purity greater than 95%. Differences were observed in the RLE and the RCP of [^99m^Tc]Tc-SFN and [^99m^Tc]Tc-FITC-SFN, indicating that FITC has an enhancement effect on the radiolabeling of NPs, contrary to ^111^In-radiolabeling of silk NPs based on chelators [10].

During the optimization of the radiolabeling, the amount of stannous chloride had to be adjusted since amounts ≥ 250 µg/mL modified the hydrodynamic properties of the SFN, producing the formation of aggregates when the quantities added were greater than 500 µg/mL. Finally, the concentration of stannous chloride of 20 μg/mL was chosen for reducing the function, while maintaining both the radiolabeling efficiency and the hydrodynamic characteristics of the NPs.

In our case, low percentages of released ^99m^Tc during the in vitro tests showed that both radiolabeled NPs, [^99m^Tc]Tc-SFN and [^99m^Tc]Tc-FITC-SFN, had high stability both in 0.9% sodium chloride and in human blood serum. In fact, the hydrodynamic characteristics of the radiolabeled NPs did not vary with statistical significance during the incubations. These results are in line with those published for the radiolabeling of SFN with ^111^In based on chelators [10], opening up the range of strategies for labeling SFN with different radionuclides depending on the biodistribution studies to be carried out.

In recent years, genotoxicity studies on nanomaterials have been increasingly used as part of the characterization of their safety profile. While there is still a lack of evidence on DNA damage related to nanomaterials, it appears that they can induce oxidative stress and trigger inflammatory responses and potentially initiate toxic responses that could be the starting point of carcinogenesis [17]. Among the in vitro methodologies available for this purpose, the CBMN assay developed by Fenech and Morley [27] has been widely applied to various types of nanomaterials, reaching heterogeneous results [17].

To allow for a thorough assessment of the type of genotoxicity experienced with SFN, CBMN was assayed in human lymphocytes scoring nuclear anomalies indicative of chromosomal instability. These anomalies were classified and expressed as chromosome aberrations and malsegregation during mitosis expressed as MNi, anaphase bridge formation expressed as NPBs, and gene amplification or elimination of unresolved DNA complexes expressed as NBUDs [28]. The samples of FITC-SFN did not generate higher frequencies of MNi, NBUDs or NPBs than those observed for the negative control (See Table 2 for details). This results showed the absence of genotoxicity related to FITC-SFN, contrary to other NPs, for instance, TiO_2_ or Ag NPs [17,29]. According to data scored, the incorporation of ^99m^Tc to the NPs did not induce statistically different genotoxicity, neither for SFN nor for FITC-SFN, compared to the negative control. NPB values showed the absence of related oxidative stress, and the NPB/MN ratio indicates, as expected, a clastogenic agent such as the ionizing radiation in the experiment with [^99m^Tc]Tc-FITC-SFN [29]. These results are important new information not only related to the knowledge of the safety profile of SFNs themselves, but also for the field of radiolabeled nanomaterials, since the proposed radiolabeling methodology, with the incorporation of ^99m^Tc in the described conditions, does not increase the risk of genotoxic effects.

Hemocompatibility assays on SFN were focused on coagulation parameters, since the interaction of NPs with blood triggers a cellular as well as a humoral reaction, with activation of platelets and induction of plasmatic coagulation [30]. Intrinsic and extrinsic pathways of plasmatic coagulation were assessed by APTT and PT, respectively. Neither APTT, PT, nor D-dimer values showed significant variations compared to the control. Our results confirmed a lack of coagulation activation or activation of fibrinolysis in the conditions assayed, while other inorganic NPs, such as coated silver NP at ~0.5 mg/mL [31], and biopolymeric NPs, such as chitosan NPs at 1 mg/mL [32], showed a significant increase in the values of APTT and coagulation activation.

LDH levels increased significantly after 4 h of incubation compared to the control, indicating a loss of cell integrity, as described by other authors [13,30]. In our case, the increase in LDH levels was found to be ~50 U/L, while other authors reported an increase of about 100 U/L, probably due to the different nature and size of the NP studied [13]. The increased LDH values were consistent with changes in complete red blood cell count, which decreased slightly after 4 h of incubation compared to controls, in line with previously published data [13,32]. According to these results, SFNs are classified as slightly hemolytic (2–5%) [32,33], unlike numerous NPs, including amorphous silica, tricalcium phosphate (TCP), hydroxyapatite (HAP), and especially silver (Ag) NPs that have been found to cause significant hemolysis, making their use in biomedical applications difficult [12]. Complementarily, C-reactive protein (CRP) is considered an important biomarker of infection and inflammation for a number of diseases [34]. The results in our experiments were in the clinical reference range (0.00–0.50 mg/dL), with no increase in CRP levels after 4 h of incubation compared to controls, and therefore without related inflammatory response.

However, a reduction in neutrophil count was assessed by flow cytometry compared to the control group after 4 h of incubation. These results are consistent with a mild activation of the inflammatory response to SFN triggered by early granulocyte activation followed by monocyte/macrophage activation [13]. In this line, some authors reported a limited inflammatory effect after 24 h of incubation with silk microparticles of 45–125 µm at 5 mg/mL [35]. Basophils, eosinophils, monocytes, and lymphocytes did not experience changes in their levels with respect to the control group, in agreement with previous studies where toxicity in lymphocytes and monocytes due to oxidative stress was studied at concentrations higher than 50 µg/mL of SFN [36].

Platelet aggregation evaluated by incubation with PRPs showed aggregation percentages in the range of 4.65–12.6%, with clear individual donor variation. These results, together with the decrease in the number of platelets after 4 h compared to the control group, suggest contact with thrombogenic material [34]; however, since the percentage of aggregation is lower than 20% [33] and the platelet count is kept in the range of reference, we could consider it as a mild thrombogenic response.

The decrease in fibrinogen and albumin levels after 4 h of incubation compared to the control indicates a slight adsorption of these proteins on the SFN surfaces, although not significant. This is the so-called Vroman effect, and it is considered the main cause of the host response to implants, including thrombogenicity and inflammation [13], due to changes in the blood interface leading to activation of the complement [33].

Therefore, we conclude that the interaction of blood with [^99m^Tc]Tc-radiolabeled SFN leads to mild activation of the inflammatory response, as described in previous studies with silk biomaterials including NPs [37], hydrogels [38], or films [39]. This response is linked to the character of the nanoparticles, and although silk fibroin is known for its anti-inflammatory activity and biocompatibility [40,41,42], the results on hemocompatibility are more related to size, surface charge, and topology of NPs, which play an important role in blood interactions regardless of the nature and composition of the silk. According to our results, SFNs could be considered safe, since the described interactions lead to mild reactions that prevent unwanted inflammation and coagulation reactions and minimize side effects such as thrombus formation [12,43]. In any case, it should be noted that in vitro hemocompatibility assays suffer from limitations mainly related to the static methodology [33,37] that to date can only be overcome with in vivo studies. Examples of in vivo hemocompatibility, such as that by Ab-del-Halim K. Y. et al. [16] showing long-term hematological effects, increase the importance of performing these tests over different and prolonged periods of time [12].

In summary, in this work, [^99m^Tc]Tc-FITC-SFN were obtained by means of a radiolabeling strategy under mild conditions, without using additional chelating agents, preserving the integrity of the NPs and without altering the original hydrodynamic properties, and hence their biodistribution. The NPs showed high in vitro stability in biological media and a safety profile characterized by the absence of genotoxic effects and a mild inflammatory response after intravenous administration. In addition, since SFNs support sterilization and freeze-drying processes [21], their formulation in a cold kit such as those for clinical use could be considered, which would lead to a reproducible and standardized procedure for labeling with ^99m^Tc.

## 4. Materials and Methods

### 4.1. Chemicals

Sodium ^[99m^Tc]pertechnetate (Na[^99m^Tc]TcO_4_) was obtained from a ^99^Mo/^99m^Tc generator (Tekcis, Curium Pharma Spain S.A. Madrid, Spain). All other chemicals and solvents used were purchased from Merck (Madrid, Spain), unless otherwise specified in the text. The ultrapure water (18.2 MΩ/cm) used in the experiments was produced in an ELGA Purelab Flex 2 (High Wycombe, UK).

### 4.2. Preparation of the Silk Fibroin Solution (SFN)

White silk cocoons were obtained from *Bombyx mori* silkworms reared in the IMIDA’s sericulture facilities (Murcia, Spain) and fed with fresh natural *Morus alba L*. leaves. Cocoons were opened by scissors and the chrysalides were removed prior to be degummed in a boiling aqueous solution of Na_2_CO_3_ 0.05 M for 120 min in order to remove the sericin efficiently and to produce smaller NPs with the highest surface charge density [44]. The silk fibroin (SF) fibers were further rinsed with ultrapure water and dried at room temperature overnight. Then, SF was dissolved at 10% (*w/v*) in 9.3M lithium bromide (LiBr) for 3 h at 65 °C, as previously described [45]. The Ambar-like silk fibroin solution was then filtered in order to remove fibers or dust particles and stored at 4 °C until use.

### 4.3. Preparation of the Silk Fibroin NPs (SFNs)

Silk fibroin NPs (SFNs) were prepared by nanoprecipitation in methanol, adapting our previously described method [10]. Briefly, the SF aqueous solution at 1% (*w/v*) was used for nanoparticle preparation by slowly dripping it into vigorously stirred methanol. After the complete addition of the SF, the nanoparticle suspension was stirred for a further 2 h to complete the transition to β-sheet, and the resulted NPs were recovered by centrifugation at 8000× *g* for 15 min at 8 °C (Eppendorf Centrifuge 5810R, Eppendorf AG, Hamburg, Germany). The pelleted particles were repeatedly washed with water in order to remove the methanol (3×). Then, the SFNs were dispersed in ultrapure water and the concentration of NPs was measured by weighting dried replicates of known volumes of the SFN suspension (n = 3). Finally, suspension was adjusted to 10 mg/mL with ultrapure water, aliquoted in vials, sterilized by autoclaving as previously described [46], and stored at 4 °C until use.

### 4.4. Fluorescent Labeling of SFNs

The FITC-labeled NPs (FITC-SFNs) were prepared by the direct reaction of a suspension of SFN in sodium carbonate buffer (160 mM, pH 9.00) with FITC dissolved in dimethyl sulfoxide (DMSO), following our previously described protocol [10]. Fluorescent NPs were sterilized by autoclave treatment at 121 °C for 30 min in a Presoclave II 80 (JP-Selecta, Barcelona, Spain) [46] and stored at 4 °C protected from light until use.

### 4.5. Radiolabeling of SFNs and FITC-SFNs with ^99m^Tc

A direct method using sodium [^99m^Tc]pertechnetate (Na[^99m^Tc]TcO_4_) from ^99^Mo/^99m^Tc generator (Tekcis, Curium Pharma Spain S.A. Madrid, Spain) and SnCl_2_ as a reducing agent was chosen. The procedure was optimized by varying the quantities of SnCl_2_ (7, 12, 20 and 250 μg) in 0.33 M HCl added to 1 mg of SFNs and ~37 MBq of Na[^99m^Tc]TcO_4_ in 1 mL of 0.9% NaCl and incubated at room temperature for 10 min.

Once the experimental conditions were optimized, aqueous suspensions of SFN and (4·10^11^ NPs/mg) mixed with 2.6 mM SnCl_2_ in 0.33 M HCl were labeled with a solution of Na[^99m^Tc]TcO_4_ (~37 MBq/mL) in pH 5.5 at room temperature for 10 min. Radiolabeled NPs were recovered by centrifugation (14,100× *g*, 10 min), and the free Na[^99m^Tc]TcO_4_ was removed by washing twice with WFI (Bbraun, Barcelona, Spain), collected, and the activity measured by using a Dose Calibrator CRC-15R (Capintec, NJ, USA). The same procedure was followed for FITC-SFN.

### 4.6. Nanoparticle Characterization

The hydrodynamic diameter of NPs and Zeta potential were determined by dynamic light scattering (DLS) using a Zetasizer Nano ZSP instrument (Malvern Instruments). All measurements were carried out using a disposable folded capillary Zeta cell containing 0.7 mL of the nanoparticle suspension in NaCl 1 mM at 20 °C, according to the manufacturer’s instructions. Zeta potential was calculated from the electrophoretic mobility of NPs using the Smoluchowski approximation. All measurements were performed in quintuplicate in 1 mM NaCl solution at 25 °C.

The morphology of the NPs was examined under electron microscopy using an FESEM APREO S (Thermo Fisher Scientific Inc., Waltham, MA, USA). Briefly, a 50 μL aliquot of an aqueous suspension of NPs (1 μg/mL) was dropped onto a clean glass wafer, dried at room temperature, and coated with 2 nm of Pt by high-vacuum sputter coating in a Leica EM ACE600 (Leica Microsystems, Barcelona, Spain). The morphology was studied by collecting images at different magnifications (15,000×, 50,000×, and 100,000×) by using a T3 detector in the immersion mode (current 50 pA, accelerating voltage of 5.00 kV, WD = 5.0 mm).

The DLS measurements and FESEM imaging of the radiolabeled NPs were performed after no detectable activity of the NPs was measured.

### 4.7. Radiolabeling Efficiency, Specific Activity, and Radiochemical Purity

Radiolabeling efficiency (RLE) can be defined as the percentage of initially added radioactivity incorporated into the NPs and can be calculated using Equation (1) [47].
(1)$$Radiolabeling\;efficiency\;\left(RLE\right)=\frac{radioactivity\;in\;particles\;}{initially\;added\;radioactivity}\times 100$$
where radioactivity was measured using a dose calibrator.

The specific activity (SA), defined as the radioactivity of a material divided by its mass [47], was measured using a dose calibrator and expressed as Bq per unit mass of [^99m^Tc]Tc-SFN. The radiochemical purity (RCP), defined as the proportion of the total radioactivity in the sample associated with the [^99m^Tc]Tc-SFN to the total radioactivity of that radionuclide present in the preparation was analyzed by instant thin-layer chromatography (iTLC) [48,49]. Five microliters of the nanoparticle suspensions was directly applied on iTLC-SG chromatographic paper (Agilent, Santa Clara, CA, USA) developed in MEK and measured in a radio-TLC scanner (AR2000, Bioscan, Washington DC, USA). Under these conditions, radiolabeled NPs and “free” ^99m^Tc were assigned with retention factors (Rf) of 0.0–0.2 and 0.8–1.0, respectively.

### 4.8. In Vitro Stability Assays

The in vitro stability of the radiolabeled NPs was evaluated in saline (0.9% NaCl) and blood serum from human healthy donors [48]. Briefly, suspensions of 35.52 ± 0.81 MBq/mg of [^99m^Tc]Tc-SFN (n = 3) or 34.04 ± 0.16 MBq/mg of [^99m^Tc]Tc-FITC-SFN (n = 3) were mixed with 2 mL of the medium and incubated for 4 h at 37 °C under gentle stirring. The stability of the radiolabeling was tested by measuring the radiochemical purity at 0, 1, and 4h of incubation. Samples from serum incubation were mixed with acetonitrile (1:1), centrifuged at 14,100× *g* for 30 min, and the supernatant was recovered for iTLC, as described in Section 4.6. Next, NPs were stored at 4 °C until no detectable activity was measured. Then, suspensions were centrifuged, and pellets were suspended at 1 mg/mL for further characterization.

### 4.9. Cytokinesis Block Micronucleus (CBMN) Assay

A volume of 5 mL of fresh venous blood (n = 3) obtained by venipuncture from healthy voluntary human donors (2 women and 1 man, aged 28 to 40 years old) was collected in 0.6 mL of acid citrate-dextrose anticoagulant A solution (Grifols S.A. Barcelona, Spain). Blood was divided into four aliquots of 1 mL and 50 µL of [^99m^Tc]Tc-FITC-SFN (~37 MBq/mg/mL) and FITC-SFNs (1 mg/mL) were added and incubated for 4 h at 37 °C. In parallel, ~37 MBq of Na[^99m^Tc]TcO_4_ (740 MBq/mL) in 1 mL of human blood and 1 mL of blood with no reagents added were incubated for 4 h at 37 °C as positive and negative controls, respectively. After incubation, the four samples underwent the CBMN assay following steps described in the next paragraph. The assay was conducted as described by Miñana et al. [50]. Briefly, 1 mL of blood containing samples (50 µg of either [^99m^Tc]Tc-FITC-SFN or FITC-SFN) or a negative control (solution of 0.9% NaCl) was added to 7 mL of a culture of lymphocytes in a selective medium (Chromosome Medium P. Euroclone S.p.A, Siziano, Italy) and cultured at 37 °C for 72 h. Then, Cytochalasin B (at a final concentration of 6 μg/mL) was added 44 h after the beginning of the culture to block cytokinesis. At the end of the incubation, the cells were harvested, treated with a hypotonic solution (0.075 M KCl) for 15 min at 37 °C [51], and fixed with a mixture of methanol/glacial acetic acid (3:1). Slides were stained with Giemsa at 5%, and micronuclei (MNi), nuclear buds (NBUDs), and nucleoplasmic bridges (NPBs) were scored at least in 1000 binucleated cytokinesis-blocked cells based on criteria summarized by Fenech [26], using a microscope with ×40 magnification objective (Olympus CX 40, Tokyo, Japan). Representative images as examples of the scored cells were taken using a cell phone adapter to fix the camera of a mobile phone (iPhone l2, Apple, CA, USA) to the objective lens.

### 4.10. Hematological Response

Tests were designed to study the hemocompatibility of the SFNs in vitro at an early and late stage by incubating whole blood with SFNs for 30 min and 4 h, respectively. The interaction with blood was characterized by measuring the following parameters: red blood cell count, platelets, neutrophils, eosinophils, basophils, lactate dehydrogenase (LDH), C-reactive Protein (CRP), albumin, prothrombin time (PT), activated partial thromboplastin time (APTT), and D-dimer.

Human whole blood from healthy volunteers (3 males and 3 females, aged 24–45 years old) was collected by venipuncture in vacutainers (BD Life Sciences, NJ USA) and exposed to SFN at a concentration of 50 µg/mL in static conditions at 37 °C for 30 min for coagulation parameters and 4 h for blood count, biochemical, and coagulation parameters. All blood donors provided informed consent.

Blood count was determined in an automated hematology analyzer (XN20, Sysmex Corporation, Kobe, Japan). Lactate dehydrogenase (LDH), C Reactive Protein (PCR), albumin and total proteins were measured in an automatic biochemistry analyzer (Cobas 8000, Roche Diagnostics, Barcelona, Spain). Proteinogram was determined by capillary electrophoresis (Capillarys 3 TERA, Sebia Hispania S.A. Barcelona Spain). Hemostasis parameters were quantified by means of ACLTOP350 (Werfen, Barcelona, Spain). Controls pre- and postincubation were included. Studies were performed by triplicate sets of samples.

To assess platelet aggregation, 1 mL of platelet-rich plasma, (PRP) obtained from blood of three healthy donors collected in sodium citrate tubes and centrifuged at 180× *g* for 15 min, was incubated with [^99m^Tc]Tc-labeled SFNs (50 µg/mL) at 37 °C for 30 min or 4 h. WFI or 0.25 M CaCl_2_ were included as negative and positive controls, respectively. Then, PRP was centrifuged at 180× *g* for 15 min, incubated (100 µL) for 5 min with 4 µL of Giemsa dye, and diluted at 1:200 with saline solution for platelet counting (PC) in a Neubauer chamber using a microscope with ×40 magnification (Olympus CX 40, Tokyo, Japan) following Jesus et al. [32]. The percentage of platelet aggregation was calculated using Equation (2).
(2)$$Platelet\;aggregation\;\left(\%\right)=\frac{PC\;negative\;control-PC\;sample\;}{PC\;negative\;control}\times 100$$
where *PC* is the platelet count.

Tests were run in the Clinical Analysis Department of HCUVA accredited by the Spanish accreditation and certification entity (ENAC), in accordance with the criteria established in the Standard ISO 15,189 (nº579/LE1193).

### 4.11. Statistical Analysis

All experiments were performed at least in triplicate and results are presented as mean value ± standard deviation (SD). Statistical analysis of the experimental results was performed by using one-way analysis of variance (ANOVA) or a paired *t*-test according the number of groups, with GraphPad Prism V.9.02 software. The reported *p*-values were considered statistically significant at *p* < 0.05 (*), *p* < 0.01 (**), *p* < 0.001 (***) and *p* < 0.0001 (****).

## 5. Conclusions

Silk fibroin nanoparticles have demonstrated chemical versatility for the integration of radioisotopes and fluorescent reporters into their structure, becoming a very attractive tool, not only for multimodal molecular imaging, but also for therapeutic purposes.

In this work, a new labeling approach for these nanoparticles leads to a dual [^99m^Tc]Tc-FITC-SFN probe for in vivo/in vitro studies. In addition to the excellent properties of ^99m^Tc itself, the strategy of incorporating the radionuclide without chelating agents and the feasibility of the pharmaceutical formulation make this approach very interesting for biodistribution studies.

## Figures and Tables

**Figure 1 pharmaceuticals-16-00248-f001:**
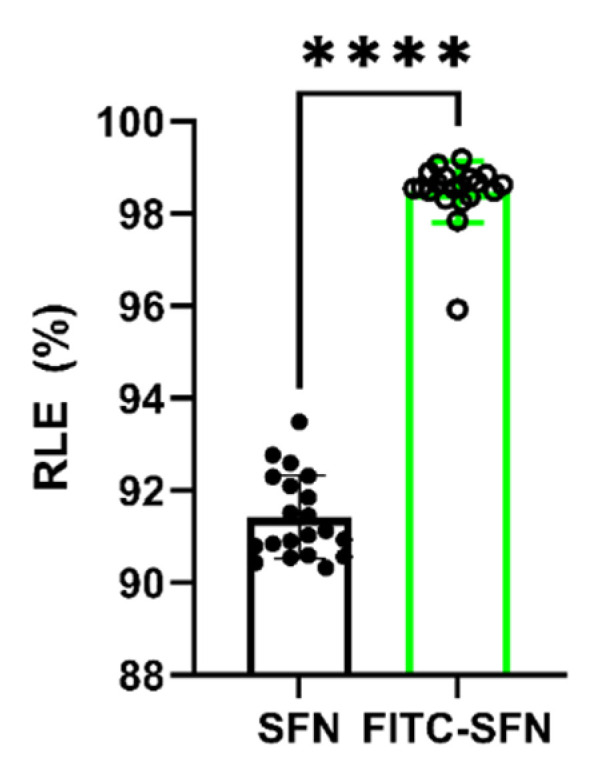
Radiolabeling efficiency (RLE) of the SFN and FITC-SFN measured in the optimized conditions. Unpaired *t*-test: ****, *p* < 0.0001; (n = 20).

**Figure 2 pharmaceuticals-16-00248-f002:**
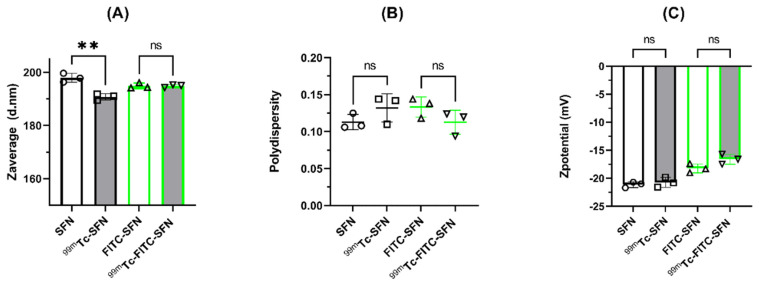
Changes in the hydrodynamic characteristics of the NPs after incubation in blood serum: (**A**) size or Z_average_ (diameter in nm); (**B**) polydispersity; and (**C**) Z_potential_ (mV). The bars or the horizontal lines represent the mean values and the whiskers the standard deviation of the mean. Overlapping symbols indicate individual measured values. Unpaired *t*-test: **, *p* < 0.01; n.s., *p* > 0.05; (n = 3).

**Figure 3 pharmaceuticals-16-00248-f003:**
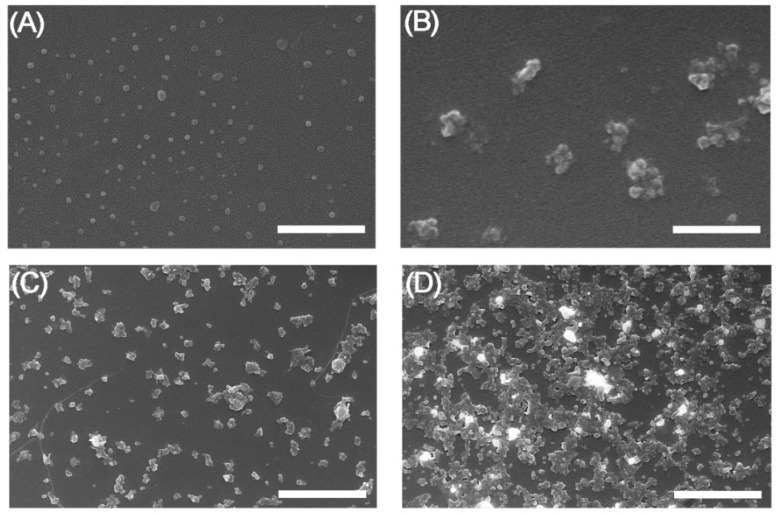
FESEM images of the NPs after incubation in serum: (**A**) SFN; (**B**) FITC-SFN; (**C**) [^99m^Tc]Tc-SFN; (**D**) [^99m^Tc]Tc-FITC-SFN. Scale bars = 500 nm, (**A**,**B**); 2 μm, (**C**,**D**).

**Figure 4 pharmaceuticals-16-00248-f004:**
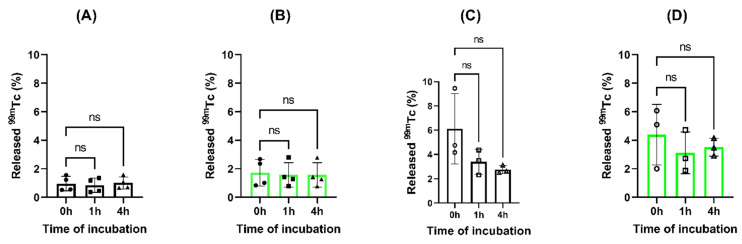
Analysis of released ^99m^Tc from the loaded nanoparticles during the time of study: (**A**) [^99m^Tc]Tc-SFN in 0.9% NaCl; (**B**) [^99m^Tc]Tc-FITC-SFN in 0.9% NaCl; (**C**) [^99m^Tc]Tc-SFN in blood serum and (**D**) [^99m^Tc]Tc-FITC-SFN in blood serum. ANOVA: n.s., *p* > 0.05; (n = 3).

**Figure 5 pharmaceuticals-16-00248-f005:**
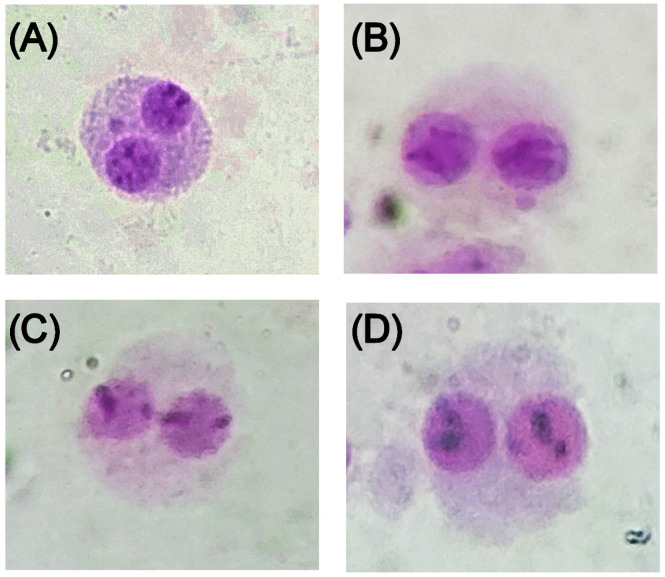
Micrographs showing binucleated cells in the lymphocyte cytokinesis-block micronucleus assay classified as (**A**) micronuclei (MNi); (**B**) nuclear buds (NBUDs); (**C**) nucleoplasmic bridges (NPBs); and (**D**) binucleated cells.

**Figure 6 pharmaceuticals-16-00248-f006:**
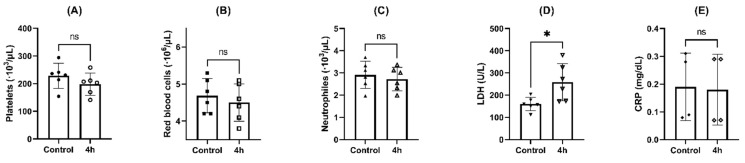
Effects of NPs on hematological parameters after incubation for 4 h at 37 °C: (**A**) platelets; (**B**) red blood cells; (**C**) neutrophils; (**D**) LDH; and (**E**) CRP. Paired *t*-test: ns = *p* > 0.05; * = *p* < 0.05, (n = 6).

**Figure 7 pharmaceuticals-16-00248-f007:**
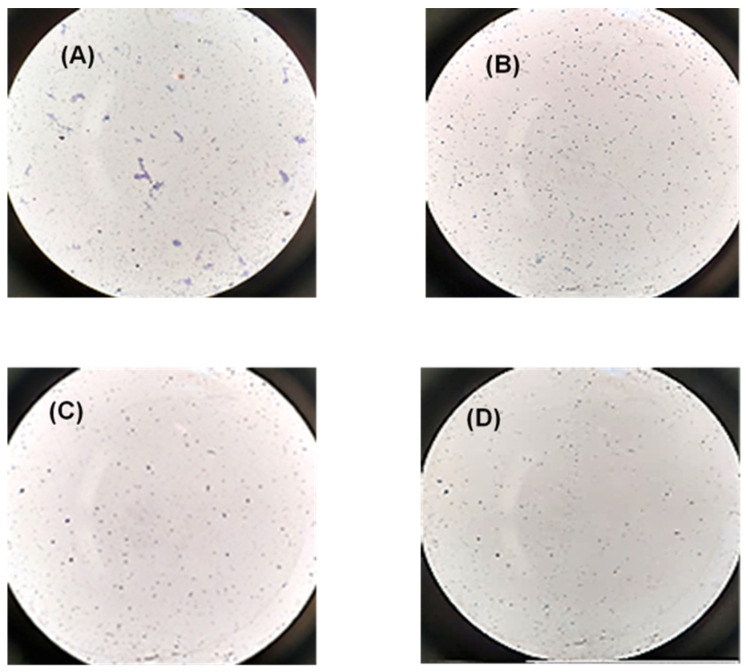
Representative microscopy images of platelet aggregation after staining with Giemsa dye in the following conditions: (**A**) PRP incubated with CaCl_2_ for 30 min (positive control); (**B**) PRP incubated with WFI (negative control); (**C**) PRP incubated with 50 µg/mL of PRP during 30 min; (**D**) PRP incubated with 50 µg/mL of PRP for 4 h. Microscope objective: 40×.

**Figure 8 pharmaceuticals-16-00248-f008:**
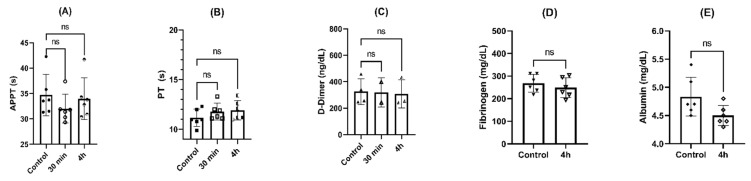
Effects of NPs on coagulation parameters after incubation for 4 h at 37 °C: (**A**) activated partial thromboplastin time (APTT); (**B**) prothrombin time; (**C**) D-dimer levels; (**D**) fibrinogen; (**E**) albumin. ns = *p* > 0.05, ANOVA or paired *t* test (n = 6).

**Table 1 pharmaceuticals-16-00248-t001:** Radiolabeling efficiency and hydrodynamic characteristics of the silk fibroin NPs as a function of SnCl_2_ during radiolabeling.

SnCl_2_ (µg/mL)	RLE (%)	Z_ave_ (d.nm)	Polydispersity	ζ (mV)
7	88.34 ± 1.11 ^a^	144.8 ± 1.6 ^a^	0.133± 0.010 ^a^	–29.5 ± 2.3 ^a^
12	89.44 ± 0.87 ^a^	144.3 ± 1.3 ^a^	0.151 ± 0.001 ^b^	–27.1 ± 1.3 ^a^
20	92.13 ± 0.96 ^b^	144.6 ± 2.6 ^a^	0.158 ± 0.016 ^b^	–26.9 ± 0.9 ^a^
250	96.38 ± 0.29 ^c^	167.1 ± 4.6 ^b^	0.258 ± 0.007 ^c^	–23.8 ± 0.5 ^b^

Values presented as mean ± standard deviation (n = 5). Different uppercase letters in the same column indicate statistically different values (*p* < 0.05).

**Table 2 pharmaceuticals-16-00248-t002:** Frequencies of micronuclei, nuclear buds, and nucleoplasmic bridges in human lymphocytes cultured in vitro with radiolabeled nanoparticles by CBMN assay.

Sample	MNi (‰)	NBUDs (‰)	NPBs (‰)
Control +	47.67 ± 9.07 ^a^	25.67 ± 2.89 ^a^	29.33 ± 8.33 ^a^
Control -	7.67 ± 4.93 ^b^	6.00 ± 2.00 ^b^	6.00 ± 3.46 ^b^
FITC-SFN	5.67 ± 2.89 ^b^	6.67 ± 3.06 ^b^	6.00 ± 1.41 ^b^
^99m^Tc-FITC-SFN	4.33 ± 2.08 ^b^	3.33 ± 0.58 ^b,c^	8.50 ± 0.71 ^b^

Values presented as mean ± standard deviation (n = 3). Different uppercase letters in the same column indicate statistically different values (*p* < 0.05, ANOVA).

**Table 3 pharmaceuticals-16-00248-t003:** Effect of SFN on platelet aggregation assessed by incubation in PRP.

	Donor #1		Donor #2		Donor #3	
Sample	Platelet Count (×10^9^/L)	Platelet Aggregation (%)	Platelet Count (×10^9^/L)	Platelet Aggregation (%)	Platelet Count (×10^9^/L)	Platelet Aggregation (%)
^†^ Control -	207.3 ± 45.5 ^a^	0.0± 20.9 ^a^	200.6 ± 40.4 ^a^	0.0± 20.1 ^a^	226.6 ± 23.2 ^a^	0.0± 10.2 ^a^
^§^ Control +	91.7 ± 10.4 ^b^	55.8 ± 5.0 ^b^	116.3 ± 12.3 ^b^	42.0 ± 6.3 ^b^	93.3 ± 16.7 ^b^	58.8 ± 7.3 ^b^
^‡^ SFN (t = 30 min)	197.3 ± 46.5 ^a^	4.8 ± 22.4 ^a^	194.6 ± 8.6 ^a^	3.0 ± 3.2 ^a^	212.6 ± 18.1 ^a^	6.2 ± 8.0 ^a^
^‡^ SFN (t = 4 h)	188.0 ± 15.9 ^a^	9.3 ± 7.6 ^a^	191.3 ± 10.3 ^a^	4.6 ± 5.1 ^a^	198.0 ± 13.1 ^a^	12.6 ± 5.8 ^a^

^†^ negative control: WFI; ^§^ positive control: 0.25M CaCl_2_; ^‡^ SFN added at 50 µg/mL. Values presented as mean ± SD (n = 3). Different uppercase letter in the same column indicates statistically different value (*p* < 0.05, ANOVA).

## Data Availability

The data presented in this study are available on request from the corresponding author.

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
