# Peer review of "In Vitro Hemocompatibility and Genotoxicity Evaluation of Dual-Labeled [^99m^Tc]Tc-FITC-Silk Fibroin Nanoparticles for Biomedical Applications"

_pharmaceuticals, 2023, doi:10.3390/ph16020248_

Round 1

Reviewer 1 Report

Dear Authors, my comments was attached as a file.

Author Response

Please see attached document with the point-by-point response to the reviewer's comments.

Reviewer 2 Report

This study very well complements the specialized literature on biodistribution studies, through the strategy of incorporating the radionuclide without chelating agents and the feasibility of pharmaceutical formulation. Nuclear imaging is a highly sensitive and non-invasive imaging technique essential for medical diagnosis, and the use of radiolabeled nanomaterials as imaging probes has shown rapid development in recent years as a tool with good sensitivity, power, and non-invasive.

This is a good manuscript on technically correct and sums up exciting results.

Line 21-22: It may be rephrasing,

Radiolabeled nanomaterials as imaging probes have shown rapid development in recent years as a powerful, highly sensitive, and non-invasive tool.

Line 24:  ….might improve the evaluation and validation …

Line 25-26: It may be rephrasing,

This work presents a direct method for [99mTc]Tc-radiolabeling of FITC-25 labeled silk fibroin nanoparticles (SFN).

Line 31-33: It may be rephrasing,

As expected, the interaction of SFN with blood provokes a mild host response, and the CBMN assay showed no genotoxicity induced by [99mTc]Tc-FITC-SFN in the conditions described.

Line 59: allows the multimodal molecular 59 imaging Space [10].

Line 123: However, at this concentration, a significant…

Pay attention on the legend of the tables and figures, they should be bolded.

Figure 5 is not good quality, you can consider improving it.

It can be accepted after minor changes.

Author Response

(The authors gave the same response as above.)
